# Effect of LMWOAs on Maize Remediation of Cadmium and Plumbum Pollution in Farmland

Ronghao Tao, Jingyi Hu, Chi Cao, Jing Zheng, Xiaotian Zhou, Hongxiang Hu, Youhua Ma 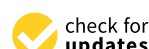, Wenling Ye, Zhongwen Ma * and Hongjuan Lu *

Anhui Province Key Laboratory of Farmland Ecological Conservation and Pollution Prevention, College of Resources and Environment, Anhui Agricultural University, Hefei 230036, China
* Correspondence: mazhongwen@ahau.edu.cn (Z.M.); hjlu@ahau.edu.cn (H.L.)

**Abstract:** Plant extraction is a thorough remediation method to remove heavy metals from soil. Chelating agents can enhance the extraction effect of heavy metals from soil by plants. In this study, low molecular weight organic acids (LMWOAs), such as citric acid (CA), tartaric acid (TA) and polyaspartate (PASP), were added to the farmland soil contaminated with Cd and Pb and combined with corn to enhance the absorption and extraction of Cadmium (Cd) and Plumbum (Pb). The effects of LMWOAs on the growth and yield of maize, Cd and Pb concentrations in each part of maize, rhizosphere soil pH, total cadmium and lead content and DTPA-Cd and Pb concentrations were studied. The enrichment, transport and extraction ability of maize were analyzed, and calcium chloride (CaCl$_2$) was compared. The results showed that: (1) Exogenous LMWOAs increased the biomass and yield of maize compared with no material added (CK), and CA increased the yield of maize by 6.33%. (2) Exogenous LMWOAs promoted the uptake of Cd and Pb in all parts of maize, and the distribution of Cd and Pb in all organs was as follows: straws > roots > maize cobs > kernels and roots > straws > maize cobs > kernels. The Cd and Pb concentrations in maize kernels were in line with GB 13078-2017 in China, which could be used as feed maize. (3) Exogenous LMWOAs enhanced the enrichment of Cd and Pb in maize straws and roots. PASP had a good enrichment effect on Cd and Pb. CA and CaCl$_2$ could enhance the transport capacity of Cd and Pb from maize roots to straw. (4) Compared with CK, the total amount of Cd and Pb in rhizosphere soil decreased by 6.93–26.99% and 2.74–6.79%, respectively. LMWOAs could promote the concentration of DTPA-Cd and Pb in rhizosphere soil, and PASP had the most significant increase in DTPA-Cd and Pb in rhizosphere soil. Compared with CK, the soil pH value decreased by 0.03–0.14 units. LMWOAs can enhance the removal of heavy metals from maize, and PASP has the most significant comprehensive effect.

**Keywords:** LMWOA; Cd-Pb pollution; maize; enrichment and extraction; remove; repair

## 1. Introduction

In recent years, with the rapid development of industrialization and agriculture, heavy metal pollution has attracted widespread attention. The development of concentrated industrial areas and agricultural production areas has produced a large number of wastes containing heavy metals and other harmful substances. In China alone, 7% of the soil has a Cd concentration exceeding the standard [1,2]. Cd has the characteristics of high mobility, good persistence and easy accumulation. It can be migrated to the environment through runoff, irrigation, sediment deposition and other ways and can also be transferred to the human body through the food chain [3,4]. Continuous exposure to Cd may cause various diseases, such as cardiovascular and cerebrovascular diseases, renal failure and nervous system problems, which seriously affect human health [5]. When agricultural land is polluted by heavy metals, this problem becomes more serious [6]. Therefore, environmental protection and food safety are the top priorities of every country.

Therefore, the remediation of heavy metal-contaminated soils has become a research hotspot for scientists around the world [7]. Heavy metals in soil are highly persistent

and nonbiodegradable and can only be converted from one chemical state to another, and the ionic form of heavy metals is more toxic than the complex. Therefore, the removal of heavy metals from agricultural soils is particularly important in order to maintain safe food chains and healthy agroecosystems. However, the large-scale removal of heavy metals from contaminated soils is a very difficult, expensive and technically demanding process [8,9]. In 1983, Chaney proposed a new concept of soil pollution remediation, which is the remediation of heavy metal-contaminated soil by using the heavy metal enrichment ability of super-enriched plants [10]. The results of engineering experiments and field applications show that phytoremediation technology has great commercial prospects. Phytoremediation methods to remove heavy metals mainly include plant extraction, stabilization and volatilization. However, phytoremediation technology also has drawbacks. For example, most super-enriched plants have slow growth, short root systems and low biomass, such as Pbs, with a time-consuming phytoremediation process and longer repair time. Moreover, the repair elements are relatively single, resulting in low efficiency. Phytoremediation is affected by environmental factors, strong specificity and a low extraction rate, such as Pb in its limited application [11,12]. Most of the heavy metals in soil exist in the form of a residue with the lowest bioavailability, which is also the main limiting factor of phytoremediation. Therefore, how to improve the efficiency of phytoremediation has become the key to the large-scale application of phytoremediation.

In order to improve the efficiency of phytoremediation, scholars have studied many combined enhancement techniques of phytoremediation in recent years, among which there are two main ways to enhance the ability of plants to extract heavy metals from soil: one is to promote the growth of super-enriched plants and improve their biomass. The second is to promote the absorption of heavy metals by plants. For example, heavy metals in soil can be activated by chelating agents, thus promoting their enrichment into plants [13,14]. A chelating agent with a soil heavy metal chelate reaction after input and forming a water-soluble metal complex increases the biological effectiveness of heavy metals, thus promoting the enrichment of plants to absorb heavy metals pollution in soil and increase the repair effect so the chelating agent is widely used during heavy metal pollution to strengthen the soil plant extract [15,16]. However, common chemical chelators can reduce soil fertility and cause secondary contamination. Studies have shown that synthetic chelators, such as EDTA, EGTA and DPTA, have great limitations in practical applications due to their environmental persistence, low biodegradability and easy-to-cause heavy metal leaching risk [17]. Therefore, it is of great significance to find a chelating agent that is environmentally friendly and efficient. Natural chelators mainly refer to low molecular weight organic acids (LMWOAs), such as citric acid, tartaric acid and oxalic acid, which are biodegradable and environmentally friendly and are considered to have strong application potential in the phytoremediation of heavy metal-contaminated soil [18]. LMWOAs are the most abundant substance involved in a heavy metal reaction, and its secretion process is an important antistress response mechanism for plants to adapt to the external environment [19]. At the same time, as an important heavy metal ligand in plants, LMWOAs are also involved in the absorption, transport, storage and other physiological metabolic processes of heavy metals [20]. Exogenous LMWOAs can chelate with Cd to form a soluble organic acid–Cd complex that can penetrate the lipid membrane of root cells, which is the main form of Cd uptake by crops [21]. LMWOAs enhanced phytoremediation mainly adopt foliar spraying and soil exogenous application. The latter can complexate with heavy metals in soil, reduce toxicity and improve the bioavailability and phytoremediation efficiency. An exogenous application of 3 mmol·kg$^{-1}$ citric acid can improve the repair index of zinc, copper, Pb, nickel and Cd of Liriope platyphylla Wang et Tang by 60–187% [22]. The treatment of 2 mmol·kg$^{-1}$ citric acid can promote the accumulation of heavy metals in the above and underground parts of Iris halophilium, and the application of organic acids can be used to repair Pb tailings to improve the mine environment [23]. The application of citric acid can improve the absorption and tolerance of Cd in ramie (*Boehmerianivea* L. Gaud.), which has a broad application prospect in the

phytoremediation of moderately Cd-polluted sites [24]. The application of acetic acid, oxalic acid, citric acid, malic acid and tartaric acid can increase the Cd absorption of Brassica napus by more than 100% [25]. The application of 10 mmol·kg$^{-1}$ citric acid can effectively improve the Cd migration coefficient of Pokeweed [26]. The combined application of citric acid and Kocuria rhizophila could increase the accumulation of Cd, chromium, copper and nickel in *Glycine max* L. by 40.63%, 56.39%, 59.1% and 39.76%, respectively.

According to GB 15618-2018 in China, edible agricultural products cannot be grown on farmlands that exceed the strictly controlled value of heavy metals in agricultural land. This will lead to a reduction in agricultural land, posing a threat to food security in China and around the world. Exogenous chelating agents can enhance the availability of heavy metals in soil and then enhance the extraction of heavy metals in plants and enrich them. From the review of previous studies, most of the materials used to enhance the extraction of heavy metals from plants are synthetic chelators. Although the extraction efficiency of heavy metals from soil is high, it is easy to cause secondary pollution to the local ecological environment due to its weak degradability in the natural environment. In addition, most of the plants used in studies are super-accumulating plants. Although they have a strong enrichment ability for heavy metals, they have defects such as a generally small biomass and cannot generate considerable economic benefits, which will reduce the production willingness of agricultural workers, and do not conform to the concept of restoration while producing. At the same time, most of the previous studies were carried out in the way of indoor pot experiments, and natural factors such as climate and precipitation under field conditions also affected the experimental results. Therefore, this study on the basis of previous research for further improvements, the three kinds of LMWOAs belong to the category of natural chelating agents, CA and TA belong to carboxylic acid compounds, and PASP belongs to compounds. The three kinds of LMWOAs are biodegradable, relatively friendly to the environment, and do not easily cause secondary pollution. In the past pot experiments, the extraction effect is significant, and there is a strong remediation potential. The field experiment was conducted from 11 June to 17 September 2021 in a cropland polluted by Cd and Pb in Tongling City, Anhui Province, China. The effects of different LMWOAs on enhancing the absorption and removal of heavy metals from soil by maize were studied, with a view toward improving the soil heavy metal pollution in China and even the world, to provide a theoretical and practical basis for alleviating the current global food security caused by heavy metal pollution.

## 2. Materials and Methods

The experimental plot is located in a tightly controlled cultivated land in Yi 'an District, Tongling City, Anhui Province, China. The concentrations of Cd and Pb in the soil are 2.41 mg·kg$^{-1}$ and 203.41 mg·kg$^{-1}$—among which, the concentrations of available Cd and Pb are 0.84 mg·kg$^{-1}$ and 54.17 mg·kg$^{-1}$, respectively. According to GB 15618-2018 in China, the heavy metal Cd concentration in the cultivated soil of the experimental field was higher than the control value. The Pb concentration was higher than the screening value but lower than the risk control value of soil pollution in agricultural land. The basic physical and chemical properties of the soil were as follows: soil pH 4.87, organic matter (OM) 19.17 g·kg$^{-1}$, total nitrogen 1.08 g·kg$^{-1}$, alkali hydrolyzable nitrogen 78.52 mg·kg$^{-1}$, available phosphorus 13.44 mg·kg$^{-1}$, and available potassium 119.36 mg·kg$^{-1}$.

Maize cultivar selection in the local suitable planting Jinju 1233.

Treatment 1: Blank control (CK): the same amount of water.

Treatment 2: Calcium chloride ($CaCl_2$): calcium chloride (111 g·mol$^{-1}$) at a concentration of 10 mmol·L$^{-1}$.

Treatment 3: Citric acid (CA): citric acid (192 g·mol$^{-1}$) at a concentration of 10 mmol·L$^{-1}$.

Treatment 4: Tartaric acid (TA): tartaric acid (150 g·mol$^{-1}$) at a concentration of 10 mmol·L$^{-1}$.

Treatment 5: Polyaspartic acid (PASP): polyaspartic acid (133 g·mol$^{-1}$) at a concentration of 10 mmol·L$^{-1}$.

The block experimental design was randomly distributed, with 3 replicates per treatment plot, a total of 30 plots, and each plot 12 m². 

The above main treatments adopted a split area treatment: Test A: Before planting maize, the soil of the planting land was activated by exogenous liquid, and the soil was turned and balanced for 1 week before maize sowing. Test B: In the ear stage of maize growth, it can activate exogenous liquid materials in rhizosphere parts of maize plants and the surrounding soil. The liquid material consumption in each test split area treatment is 64 L.

According to the local high-yield cultivation techniques, the base fertilizer is 45% (15-15-15) NPK compound fertilizer, with the application rate of 600 kg per hectare, applied 1–2 days before planting. In the trumpet mouth stage, 375 kg urea per hectare should be applied as topdressing, and local irrigation water should be used to combine water and fertilizer. According to the local planting habits, we adopted double-row planting, surrounded by a protection row. On 11 June 2021, the plant was directly planted on demand, and normal field water and fertilizer management and pest and grass control were carried out. On 17 September, the mature stage was sampled and harvested.

The maize plant samples were randomly sampled, and 3–5 whole maize samples with uniform growth were collected from each plot. After collection, the maize plant samples in each plot were washed with tap water and deionized water, and then, the whole plant was divided into roots, straws, maize cobs, and kernels to form mixed samples. The plants were killed at 105 °C for 30 min and dried at 80 °C to a constant weight. The dry mass of each part was weighed and crushed in a stainless steel mill. The rhizosphere soil samples were collected on the day of maize sample collection, and the corresponding rhizosphere soil samples (0–20 cm) were collected with a wooden shovel at the point where maize samples were taken to form a mixed soil sample. After air drying in the shade, pulverize and grind 10 mesh sieving and 100 mesh sieving into Ziplock bags for use.

According to Chinese GB/T 23739-2009, the DTPA-Cd and DTPA-Pb in soil were determined by the Jena Z700P atomic absorption spectrophotometer flame method. The concentration of heavy metals Cd and Pb in various parts of maize was determined according to GB 5009.268-2016 in China and was determined by the Z700P atomic absorption spectrophotometer in Jena, Germany. Soil pH was extracted, with $CO_2$ removed from distilled water (soil–water ratio 1:2.5) and measured with a precision pH meter (TARTER 2100). N, P, K, and other indexes in soil were determined by the method specified in the soil agrochemical analysis. Soil sample (GBW 07461) and plant sample (GBW 10045) were used for quality control, and the analysis results were within the allowable error range.

Relevant indicators were calculated according to the following formula:

$$\text{BCF} = \frac{\text{Element concentration in plant } \left(\text{mg·kg}^{-1}\right)}{\text{Element concentration in soil}\left(\text{mg·kg}^{-1}\right)} \tag{1}$$

$$\text{TF} = \frac{\text{Element concentration in plant } \left(\text{mg·kg}^{-1}\right)}{\text{Element concentration in plant }\left(\text{mg·kg}^{-1}\right)} \tag{2}$$

$$\text{EA (Cd)} = Q_{\text{Root}} \times C_{\text{Cd-Root}} + Q_{\text{Stem}} \times C_{\text{Cd-Stem}} + Q_{\text{Leaf}} \times C_{\text{Cd-Maize cob}} + Q_{\text{Kernel}} \times C_{\text{Cd-Kernel}}$$
$$\text{EA (Pb)} = Q_{\text{Root}} \times C_{\text{Pb-Root}} + Q_{\text{Stem}} \times C_{\text{Pb-Stem}} + Q_{\text{Leaf}} \times C_{\text{Pb-Maize cob}} + Q_{\text{Kernel}} \times C_{\text{Pb-Kernel}} \tag{3}$$

$$\text{Estimated extraction amount per hectare} = \text{Amount of heavy metals extracted from plants} \times \text{Number of trees per hectare} \tag{4}$$

$$\text{Extraction efficiency} = \frac{\text{Amount of heavy metals extracted from plants}}{\left(\text{Concentration of heavy metals in soil } \times \text{ Soil quality}\right) \times 100\%} \tag{5}$$

Excel 2016 was used for data sorting, SPPS 23.0 was used for analysis of variance, and Origin 2017 was used for mapping. Data were expressed as the mean ± error, and significant differences were tested by Duncan's test ($p < 0.05$).

## 3. Results

### 3.1. Effects of LMWOAs on Maize Biomass, Yield, and Physiological Indexes

As can be seen from Table 1, exogenous LMWOAs can improve the biomass of maize plants to a certain extent compared with CK. The biomass of maize roots, straws, cobs, and kernels were 4.60–4.68, 42.45–45.08, 25.63–26.77 g·plant$^{-1}$, and 152.7–155.9 g·plant$^{-1}$ when activated material was added before maize planting. Compared with the blank control, the maize yield ranged from 9.24 t·hm$^{-2}$ to 9.35 t·hm$^{-2}$, and the yield increase ranged from 0.99–2.37%. The plant height, leaf length, and leaf width of mature maize ranged from 207.7 to 229.0 cm, 67.33 to 91.77 cm, and 8.80 to 9.57 cm, respectively.

**Table 1.** Effects of LMWOAS on maize biomass, yield, and physiological indexes.

| Test | Treatments | Maize Biomass (g·Plant$^{-1}$) | | | | Yield (t·hm$^{-2}$) | Plant Height (cm) | Leaf Length (cm) | Leaf Width (cm) |
|------|-----------|------|-------|-----|--------|-------|--------------|-------------|------------|
| | | Root | Straw | Cob | Kernel | | | | |
| A | CK | 4.670 ± 0.090 b | 43.37 ± 1.459 a | 26.12 ± 2.440 a | 152.2 ± 3.660 a | 9.133 ± 0.217 c | 222.9 ± 8.281 ab | 70.83 ± 4.517 bc | 8.900 ± 0.557 a |
| | CA | 4.677 ± 0.085 b | 42.78 ± 3.440 a | 26.77 ± 2.523 a | 155.9 ± 2.916 ab | 9.350 ± 0.171 abc | 218.0 ± 17.49 ab | 91.77 ± 8.893 a | 8.833 ± 0.321 a |
| | CaCl$_2$ | 4.670 ± 0.056 b | 44.09 ± 2.455 a | 27.49 ± 2.735 a | 153.7 ± 2.581 ab | 9.223 ± 0.155 bc | 207.7 ± 9.016 ab | 75.07 ± 7.332 abc | 9.567 ± 0.681 a |
| | TA | 4.647 ± 0.060 b | 45.08 ± 2.315 a | 26.55 ± 2.359 a | 152.7 ± 5.633 b | 9.263 ± 0.234 bc | 209.1 ± 9.028 ab | 84.40 ± 7.825 abc | 8.800 ± 0.781 a |
| | PASP | 4.600 ± 0.089 b | 42.45 ±3.452 a | 25.63 ± 1.539 a | 152.7 ± 3.505 b | 9.240 ± 0.128 bc | 229.0 ± 11.17 a | 80.37 ± 0.850 abc | 9.100 ± 1.411 a |
| B | CK | 4.690 ± 0.089 b | 44.39 ± 1.842 a | 27.06 ± 1.330 a | 154.3 ± 3.025 ab | 9.160 ± 0.050 c | 222.9 ± 3.707 ab | 70.67 ± 2.967 bc | 8.833 ± 0.306 a |
| | CA | 4.950 ± 0.079 a | 45.28 ± 3.055 a | 28.87 ± 1.064 a | 159.6 ± 2.116 ab | 9.580 ± 0.128 abc | 209.3 ± 13.68 ab | 82.90 ± 10.95 abc | 9.000 ± 0.458 a |
| | CaCl$_2$ | 5.013 ± 0.047 a | 48.10 ± 1.545 a | 29.62 ± 2.047 a | 162.3 ± 2.666 a | 9.740 ± 0.161 a | 202.6 ± 7.399 b | 90.07 ± 9.019 ab | 7.833 ± 0.404 a |
| | TA | 4.913 ± 0.047 a | 46.64 ± 2.536 a | 28.71 ± 1.471 a | 160.8 ± 3.263 ab | 9.653 ± 0.193 ab | 203.4 ± 10.32 b | 67.33 ± 0.643 c | 8.633 ± 0.681 a |
| | PASP | 4.953 ± 0.070 a | 46.33 ± 2.025 a | 28.01 ± 1.488 a | 157.7 ± 1.878 ab | 9.463 ± 0.110 abc | 202.0 ± 13.43 b | 85.63 ± 9.448 abc | 8.433 ± 0.379 a |

Note: Different lowercase letters in the same column indicate a significant difference between treatments ($p < 0.05$).

The biomass of maize roots, straws, cobs, and kernels were 4.91–5.01, 45.28–48.10, 28.01–29.62 g·plant$^{-1}$, and 157.68-162.28 g·plant$^{-1}$ in the rhizosphere of maize at the ear stage. Compared with CK, the yield of maize ranged from 9.46 to 9.74 t·hm$^{-2}$, and the yield ranged from 3.31% to 6.33%. The plant height, leaf length, and leaf width of mature maize were 202.0–209.3, 67.33–90.07, and 7.83–9.00 cm, respectively.

### 3.2. Effects of LMWOAs on Absorption, Enrichment, and Transport of Heavy Metals in Maize

3.2.1. Effect of LMWOAs on Cd and Pb Concentration in Maize Grains

As shown in Figure 1, compared with CK, different treatments promoted the absorption of Cd and Pb in maize kernels. In test A, the mass fractions of Cd and Pb in maize kernels varied between 0.221 and 0.250 mg·kg$^{-1}$ and 0.425 and 0.471 mg·kg$^{-1}$ under different treatments, respectively. Compared with CK, the mass fractions of Cd and Pb in maize kernels were increased by 9.26–23.91% and 3.40–14.44%. Exogenous CaCl$_2$ and CA had the most significant effect on improving the quality fractions of Cd and Pb in maize kernels, followed by exogenous PASP and TA, respectively. In test B, the mass fractions of Cd and Pb in kernels varied between 0.251 and 0.314 mg·kg$^{-1}$ and 0.430 and 0.550 mg·kg$^{-1}$ under different treatments. Compared with CK, the mass fractions of Cd and Pb in maize kernels were increased by 23.04–53.92% and 5.39–34.73%, respectively. Exogenous TA and CA

had the most significant effect on the improvement of the mass fractions of Cd and Pb in maize kernels. Although the concentration of Cd and Pb in maize kernels under different treatments exceeded the limit values stipulated for GB 2762-2017 in China, they met the limit values of Cd and Pb stipulated for GB 13078-2017 in China and could be used as raw materials for plant feed.

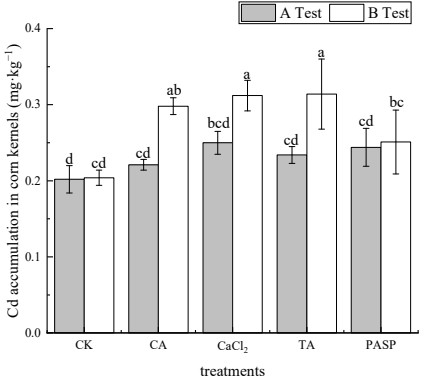 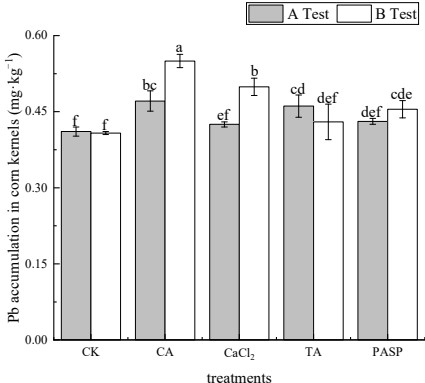

**Figure 1.** Effect of LMWOAs on Cd and Pb concentrations in maize kernels. Different small letters above the same column indicate significant difference between treatments ($p < 0.05$).

### 3.2.2. Effect of LMWOAs on Cd Concentration in Different Parts of Maize

As shown in Table 2, application of the activator could promote the uptake of Cd in all parts of maize plants, and the Cd concentration in all parts of maize plants in all treatments was as follows: straws > roots > maize cobs. In test A, the mass fraction of Cd in maize cobs, straws, and roots varied from 0.280 to 0.320, 1.079 to 1.359 mg·kg$^{-1}$, and 0.937 to 1.312 mg·kg$^{-1}$, respectively. Compared with CK, exogenous CA increased the mass fraction of Cd in stems and roots by 47.26% and 33.65%, respectively, while decreasing the mass fraction of Cd in maize cobs by 2.74%. Exogenous CaCl$_2$ increased the mass fraction of Cd in stems and roots by 61.79% and 42.88%, respectively, while decreasing the mass fraction of Cd in maize cobs by 4.31%. Exogenous TA increased the mass fraction of Cd in maize cobs, straws, and roots by 8.11%, 48.49%, and 86.33%, respectively. Exogenous PASP increased the mass fraction of Cd in maize cobs, straws, and roots by 9.48%, 28.49%, and 87.08%, respectively.

**Table 2.** Effects of LMWOAS on the Cd concentration in different parts of maize.

| Test | Treatments | Maize Cob Cd (mg·kg$^{-1}$) | Straw Cd (mg·kg$^{-1}$) | Root Cd (mg·kg$^{-1}$) |
|------|-----------|------------------------------|---------------------------|-------------------------|
|   | CK | $0.292 \pm 0.009$ ab | $0.840 \pm 0.049$ c | $0.701 \pm 0.023$ e |
|   | CA | $0.284 \pm 0.012$ b | $1.237 \pm 0.059$ ab | $0.937 \pm 0.093$ d |
| A | CaCl$_2$ | $0.280 \pm 0.017$ b | $1.359 \pm 0.218$ a | $1.002 \pm 0.072$ d |
|   | TA | $0.316 \pm 0.022$ a | $1.247 \pm 0.238$ ab | $1.307 \pm 0.012$ b |
|   | PASP | $0.320 \pm 0.018$ a | $1.079 \pm 0.059$ bc | $1.312 \pm 0.027$ b |
|   | CK | $0.282 \pm 0.012$ b | $0.839 \pm 0.047$ c | $0.716 \pm 0.006$ e |
|   | CA | $0.299 \pm 0.019$ ab | $1.316 \pm 0.074$ ab | $0.964 \pm 0.049$ d |
| B | CaCl$_2$ | $0.303 \pm 0.005$ ab | $1.219 \pm 0.076$ ab | $1.184 \pm 0.111$ c |
|   | TA | $0.283 \pm 0.003$ b | $1.149 \pm 0.145$ ab | $1.468 \pm 0.039$ a |
|   | PASP | $0.305 \pm 0.014$ ab | $1.162 \pm 0.042$ ab | $1.584 \pm 0.052$ a |

Note: Different lowercase letters in the same column indicate a significant difference between treatments ($p < 0.05$).

In test B, the mass fraction of Cd in maize cobs, straws, and roots varied from 0.283 to 0.305, 1.149 to 1.316 mg·kg$^{-1}$, and 0.964 to 1.584 mg·kg$^{-1}$, respectively. Compared with CK, exogenous CA increased the mass fraction of Cd in maize cobs, straws, and roots by 6.03%, 56.85%, and 34.54%, respectively. Exogenous CaCl$_2$ increased the mass fraction of

Cd in maize cobs, straws, and roots by 7.56%, 45.29%, and 65.34%, respectively. Exogenous TA increased the mass fraction of Cd in maize cobs, straws, and roots by 0.35%, 36.91%, and 104.94%, respectively. Exogenous PASP increased the mass fraction of Cd in maize cobs, straws, and roots by 8.38%, 38.53%, and 121.18%, respectively.

### 3.2.3. Effect of LMWOAs on Pb Concentration in Different Parts of Maize

According to Table 3, application of an activator could promote the absorption of Pb in all parts of maize plants, and the Pb concentration in all parts of maize plants in all treatments was as follows: roots > straws > maize cobs. In test A, the mass fraction of Pb in maize cobs, straws, and roots varied from 2.528 to 2.794, 39.31 to 48.88 mg·kg$^{-1}$, and 59.07 to 74.83 mg·kg$^{-1}$, respectively. Compared with CK, exogenous CA increased the mass fraction of Pb in maize cob, stem, and leaf and root by 4.60%, 24.56%, and 13.06%, respectively. Exogenous CaCl$_2$ increased the mass fraction of Pb in maize cob, stem, and leaf and root by 15.61%, 41.33%, and 10.47%, respectively. Exogenous TA increased the mass fraction of Pb in maize cob, stem, and leaf and root by 9.07%, 54.86%, and 39.94%, respectively. Exogenous PASP increased the concentration of Pb in maize cobs, stems, and roots by 13.54%, 46.33%, and 38.72%, respectively.

**Table 3.** Effects of LMWOAs on the Pb concentration in different parts of maize.

| Test | Treatments | Maize Cob Pb (mg·kg$^{-1}$) | Straw Pb (mg·kg$^{-1}$) | Root Pb (mg·kg$^{-1}$) |
|---|---|---|---|---|
| A | CK | 2.417 ± 0.113 e | 31.56 ± 0.980 e | 53.48 ± 5.135 ef |
| | CA | 2.528 ± 0.051 de | 39.31 ± 1.289 d | 60.46 ± 3.072 de |
| | CaCl$_2$ | 2.794 ± 0.110 bc | 44.61 ± 3.424 c | 59.07 ± 4.835 e |
| | TA | 2.636 ± 0.147 cd | 48.88 ± 1.201 ab | 74.83 ± 2.709 b |
| | PASP | 2.744 ± 0.030 bc | 46.18 ± 0.721 bc | 74.18 ± 2.758 bc |
| B | CK | 2.368 ± 0.016 e | 32.25 ± 0.447 e | 51.26 ± 0.298 f |
| | CA | 3.033 ± 0.098 a | 41.28 ± 1.077 d | 66.83 ± 1.020 cd |
| | CaCl$_2$ | 2.907 ± 0.035 ab | 48.30 ± 0.698 b | 72.01 ± 1.853 bc |
| | TA | 2.714 ± 0.079 c | 52.05 ± 1.877 a | 78.69 ± 4.971 b |
| | PASP | 2.653 ± 0.020 cd | 48.89 ± 1.938 ab | 86.63 ± 5.560 a |

Note: Different lowercase letters in the same column indicate a significant difference between treatments ($p < 0.05$).

In test B, the mass fraction of Pb in maize cobs, straws, and roots varied from 2.653 to 3.033, 41.28 to 52.05 mg·kg$^{-1}$, and 66.83 to 86.63 mg·kg$^{-1}$, respectively. Compared with CK, exogenous CA increased the mass fraction of Pb in maize cobs, straws, and roots by 28.09%, 27.99%, and 30.37%, respectively. Exogenous CaCl$_2$ increased the mass fraction of Pb in maize cobs, straws, and roots by 22.79%, 49.75%, and 40.48%, respectively. Exogenous TA increased the mass fraction of Pb in maize cob, stem, and leaf and root by 14.63%, 61.39%, and 53.52%, respectively. Exogenous PASP increased the concentration of Pb in maize cobs, straws, and roots by 12.05%, 51.58%, and 69.01%, respectively.

### 3.2.4. Effects of LMWOAs on Cd Enrichment and Transport in Different Parts of Maize

According to Table 4, LMWOAs can improve the ability of maize plants to enrich and transport heavy metal Cd. In test A, under different treatments, the enrichment coefficients of straw Cd ranged from 0.448 to 0.564, and those of root Cd ranged from 0.389 to 0.544, which increased by 28.39–68.74% and 33.68–87.04%, respectively, compared with CK. CaCl$_2$ and PASP had the strongest ability to enrich Cd in straws and roots. The transport coefficients of Cd from straw–roots and kernels–straw were 0.823–1.356 and 0.212–0.267, respectively. In test B, under different treatments, the enrichment coefficients of straw Cd ranged from 0.476 to 0.546, and those of root Cd ranged from 0.400 to 0.657, which were increased by 36.75–56.76% and 34.54–121.09%, respectively, compared with CK. CA and PASP had the strongest ability to enrich Cd in straws and roots. The transfer

coefficients of cadmium from straw–roots and kernels–straw were 0.734–1.370 and 0.252–0.297, respectively.

**Table 4.** Effects of LMWOAS on Cd enrichment and transport in different parts of maize.

| Test | Treatments | $BCF_{Straw}$ | $BCF_{Root}$ | $TF_{Straw/Root}$ | $TF_{Kernel/Straw}$ |
|------|-----------|--------------|--------------|-------------------|---------------------|
|      | CK | 0.349 ± 0.021 b | 0.291 ± 0.010 d | 1.200 ± 0.101 ab | 0.327 ± 0.018 a |
|      | CA | 0.513 ± 0.024 a | 0.389 ± 0.038 c | 1.332 ± 0.198 a | 0.254 ± 0.007 bc |
| A    | CaCl$_2$ | 0.564 ± 0.091 a | 0.416 ± 0.030 c | 1.356 ± 0.196 a | 0.212 ± 0.037 c |
|      | TA | 0.517 ± 0.099 a | 0.542 ± 0.005 b | 0.955 ± 0.183 cd | 0.254 ± 0.060 bc |
|      | PASP | 0.448 ± 0.025 ab | 0.544 ± 0.011 b | 0.823 ± 0.056 cd | 0.267 ± 0.017 bc |
|      | CK | 0.348 ± 0.019 b | 0.297 ± 0.003 d | 1.171 ± 0.070 ab | 0.325 ± 0.019 a |
|      | CA | 0.546 ± 0.031 a | 0.400 ± 0.020 c | 1.370 ± 0.139 a | 0.297 ± 0.020 ab |
| B    | CaCl$_2$ | 0.506 ± 0.032 a | 0.491 ± 0.046 b | 1.036 ± 0.124 bc | 0.274 ± 0.022 ab |
|      | TA | 0.476 ± 0.060 ab | 0.609 ± 0.016 a | 0.781 ± 0.085 cd | 0.252 ± 0.033 bc |
|      | PASP | 0.482 ± 0.018 ab | 0.657 ± 0.022 a | 0.734 ± 0.036 d | 0.261 ± 0.000 bc |

Note: Different lowercase letters in the same column indicate a significant difference between treatments ($p < 0.05$).

3.2.5. Effects of LMWOAs on Pb enrichment and Transport in Different Parts of Maize

According to Table 5, LMWOAs can improve the ability of maize plants to enrich and transport heavy metal Pb. In testA, under different treatments, the enrichment coefficients of straw and root Pb were 0.193–0.240 and 0.291–0.368, respectively, which increased by 24.47–54.73% and 10.66–40.08% compared with CK. TA had the strongest ability to enrich Pb in both straws and roots, and the transport coefficients of cadmium between straw–roots and kernels–straw were 0.623–0.755 and 0.009–0.012, respectively. In test B, under different treatments, the enrichment coefficients of straw Pb ranged from 0.203 to 0.256, and those of root Pb ranged from 0.328 to 0.426, which were increased by 10.66–40.08% and 30.12–68.73%, respectively, compared with CK. TA and PASP had the strongest ability to enrich Pb in straws and roots, respectively. The transport coefficients of Pb in straw–roots and kernels–straw ranged from 0.566 to 0.671 and 0.008 to 0.013, respectively.

**Table 5.** Effects of LMWOAs on Pb enrichment and transport in different parts of maize.

| Test | Treatments | $BCF_{Straw}$ | $BCF_{Root}$ | $TF_{Straw/Root}$ | $TF_{Kernel/Straw}$ |
|------|-----------|--------------|--------------|-------------------|---------------------|
|      | CK | 0.155 ± 0.005 e | 0.263 ± 0.025 e | 0.593 ± 0.050 cd | 0.013 ± 0.001 a |
|      | CA | 0.193 ± 0.007 d | 0.297 ± 0.015 de | 0.651 ± 0.016 b | 0.012 ± 0.001 a |
| A    | CaCl$_2$ | 0.219 ± 0.017 c | 0.291 ± 0.024 de | 0.755 ± 0.007 a | 0.009 ± 0.001 bc |
|      | TA | 0.240 ± 0.006 ab | 0.368 ± 0.013 bc | 0.654 ± 0.031 b | 0.009 ± 0.001 bc |
|      | PASP | 0.227 ± 0.004 bc | 0.365 ± 0.014 bc | 0.623 ± 0.028 bc | 0.009 ± 0.000 bc |
|      | CK | 0.159 ± 0.002 e | 0.252 ± 0.002 e | 0.629 ± 0.006 bc | 0.013 ± 0.000 a |
|      | CA | 0.203 ± 0.005 d | 0.328 ± 0.005 cd | 0.618 ± 0.022 bcd | 0.013 ± 0.001 a |
| B    | CaCl$_2$ | 0.237 ± 0.003 b | 0.354 ± 0.009 bc | 0.671 ± 0.012 b | 0.010 ± 0.001 b |
|      | TA | 0.256 ± 0.009 a | 0.387 ± 0.024 ab | 0.663 ± 0.032 b | 0.008 ± 0.001 c |
|      | PASP | 0.240 ± 0.009 ab | 0.426 ± 0.027 a | 0.566 ± 0.041 d | 0.009 ± 0.001 bc |

Note: Different lowercase letters in the same column indicate a significant difference between treatments ($p < 0.05$).

*3.3. Remediation Capacity of LMWOAs for Contaminated Soil*

3.3.1. Remediation Capacity of LMWOAs for Cd Contaminated Soil

Exogenous LMWOAs can reduce the concentration of heavy metals in soil, promote the absorption of heavy metals in soil by maize plants, and improve the remediation ability of polluted soil. It can be seen from Table 6 that LMWOAs have different remediation abilities for Cd-contaminated soil. In test A, the extraction amount of Cd per maize plant under different treatments ranged from 0.099 to 0.111 mg·plant$^{-1}$, which was increased by 24.53–42.31% compared with CK. CaCl$_2$ had the strongest ability to extract heavy metal

Cd from maize plants. The extraction amount of $CaCl_2$ (6.667 g·hm$^{-2}$) was the highest, followed by TA (6.360 g·hm$^{-2}$). According to the calculation of repairing the 20 cm surface soil, the average Cd extraction efficiency of each treatment was 0.117%—among which, $CaCl_2$ and PASP were the highest, which were 0.132% and 0.120%, respectively.

**Table 6.** Remediation capacity of LMWOAs for Cd-contaminated soil.

| Test | Treatments | EA (mg·Plant$^{-1}$) | Extracting Amount (g·hm$^{-2}$) | Extraction Yield % |
|------|------------|---------------------|-------------------------------|---------------------|
| A | CK | 0.078 ± 0.005 d | 4.683 ± 0.244 d | 0.073 ± 0.008 c |
| | CA | 0.099 ± 0.007 c | 5.963 ± 0.391 c | 0.100 ± 0.008 bc |
| | $CaCl_2$ | 0.111 ± 0.013 abc | 6.667 ± 0.769 abc | 0.132 ± 0.018 ab |
| | TA | 0.106 ± 0.009 bc | 6.360 ± 0.515 bc | 0.117 ± 0.016 b |
| | PASP | 0.097 ± 0.007 c | 5.833 ± 0.455 c | 0.120 ± 0.014 b |
| B | CK | 0.080 ± 0.003 d | 4.787 ± 0.201 d | 0.074 ± 0.006 c |
| | CA | 0.120 ± 0.002 ab | 7.223 ± 0.162 ab | 0.136 ± 0.011 ab |
| | $CaCl_2$ | 0.124 ± 0.008 a | 7.453 ± 0.531 a | 0.159 ± 0.016 a |
| | TA | 0.120 ± 0.007 ab | 7.177 ± 0.428 ab | 0.134 ± 0.017 ab |
| | PASP | 0.110 ± 0.006 abc | 6.583 ± 0.365 abc | 0.125 ± 0.011 ab |

Note: Different lowercase letters in the same column indicate a significant difference between treatments ($p < 0.05$).

In test B, the extraction amount of Cd per plant of maize varied from 0.110 to 0.1248 mg·plant$^{-1}$, which was increased by 37.53–55.68% compared with CK. $CaCl_2$ had the strongest ability to extract heavy metal Cd from maize plants. The extraction amount of $CaCl_2$ (7.453 g·hm$^{-2}$) was the highest, followed by CA (7.223 g·hm$^{-2}$). According to the calculation of repairing 20 cm surface soil, the average extraction efficiency of Cd in each treatment is 0.138%—among which, $CaCl_2$ and CA are the highest, which are 0.159% and 0.136%, respectively.

### 3.3.2. Remediation Capacity of LMWOAs for Pb-Contaminated Soil

It can be seen from Table 7 that LMWOAs have different remediation abilities for Pb-contaminated soil. In test A, the extraction amount of Pb per maize plant was 2.273–2.913 mg·plant$^{-1}$ under different treatments, which was increased by 20.92–54.95% compared with CK. TA had the strongest ability to extract heavy metal Pb from maize plants. The extraction amount of TA (174.8 g·hm$^{-2}$) was the highest, followed by PASP (158.2 g·hm$^{-2}$). According to the calculation of repairing the 20 cm surface soil, the average Pb extraction efficiency of each treatment was 0.029%—among which, $CaCl_2$ and PASP were the highest at 0.033% and 0.031%, respectively.

**Table 7.** Remediation capacity of LMWOAs in Pb-contaminated soil.

| Test | Treatments | EA (mg·Plant$^{-1}$) | Extracting Amount (g·hm$^{-2}$) | Extraction Yield % |
|------|------------|---------------------|-------------------------------|---------------------|
| A | CK | 1.880 ± 0.039 e | 122.8 ± 2.360 a | 0.021 ± 0.001 d |
| | CA | 2.273 ± 0.161 d | 136.4 ± 9.700 d | 0.026 ± 0.002 c |
| | $CaCl_2$ | 2.578 ± 0.185 cd | 154.7 ± 11.10 cd | 0.030 ± 0.002 b |
| | TA | 2.913 ± 0.189 ab | 174.8 ± 11.34 ab | 0.033 ± 0.002 ab |
| | PASP | 2.636 ± 0.209 bc | 158.2 ± 12.49 bc | 0.031 ± 0.003 b |
| B | CK | 1.942 ± 0.040 e | 116.5 ± 2.396 a | 0.021 ± 0.001 d |
| | CA | 2.564 ± 0.175 cd | 153.8 ± 10.50 cd | 0.030 ± 0.002 b |
| | $CaCl_2$ | 3.084 ± 0.125 a | 185.1 ± 7.482 a | 0.036 ± 0.001 a |
| | TA | 3.202 ± 0.140 a | 192.1 ± 8.419 a | 0.037 ± 0.002 a |
| | PASP | 3.068 ± 0.155 a | 184.0 ± 9.296 a | 0.037 ± 0.002 a |

Note: Different lowercase letters in the same column indicate a significant difference between treatments ($p < 0.05$).

In test B, the extraction amount of Pb per plant of maize varied from 2.564 to 3.202 mg·plant$^{-1}$, which was 32.05–64.89% higher than that of CK. TA had the strongest ability to extract heavy metal Pb from maize plants. The extraction amount of TA (192.1 g·hm$^{-2}$) was the highest, followed by CaCl$_2$ (185.1 g·hm$^{-2}$). According to the calculation of repairing the 20 cm surface soil, the average Pb extraction efficiency of each treatment was 0.035%—among which, TA and PASP were the highest, both of which were 0.037%.

### 3.4. Effects of LMWOAs on Heavy Metals in Rhizosphere Soil of Maize at Maturity Stage

#### 3.4.1. Effects of LMWOAs on Total Heavy Metals Cd and Pb in Rhizosphere Soil of Maize at Maturity Stage

According to Figure 2, LMWOAs can enhance the absorption capacity of maize plants to heavy metals Cd and Pb in soil. The total amount of Cd and Pb in rhizosphere soil decreased in different degrees under different treatments. In test A, the mass fractions of Cd and Pb in rhizosphere soil under different treatments were 1.084–2.226 mg·kg$^{-1}$ and 190.72–196.61 mg·kg$^{-1}$, which decreased by 6.93–24.59% and 2.74–5.65% compared with CK, respectively. PASP had the most significant reduction in total Cd and Pb. In test B, the mass fractions of Cd and Pb in the rhizosphere soil under different treatments were 1.743–1.987 mg·kg$^{-1}$ and 186.72–193.52 mg·kg$^{-1}$, respectively. Compared with CK, it decreased by 16.74–26.99% and 3.39–6.79%, respectively. PASP and CaCl$_2$ had the most significant reductions in total Cd and Pb, respectively.

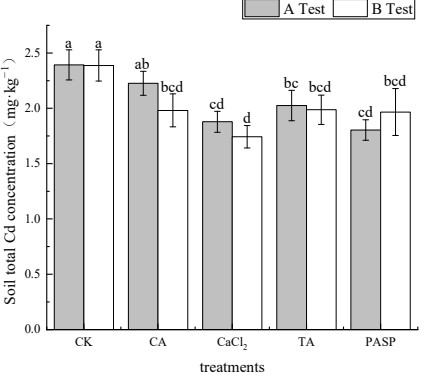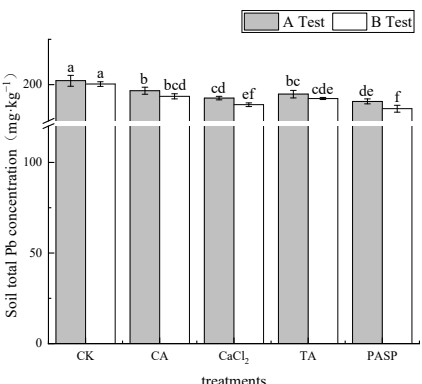

**Figure 2.** Effects of LMWOAs on the total Cd and Pb concentration in rhizosphere soil of maize at the maturity stage. Different small letters above the same column indicate significant difference between treatments ($p < 0.05$).

#### 3.4.2. Effects of LMWOAs on Heavy Metals DTPA-Cd and Pb in Rhizosphere Soil of Maize at Maturity Stage

According to Figure 3, compared with the blank, the activities of available soil heavy metals Cd and Pb were promoted by different treatments. In test A, the mass fractions of available Cd and Pb in rhizosphere soil were 1.041–1.175 mg·kg$^{-1}$ and 77.12–80.30 mg·kg$^{-1}$, respectively, which increased by 28.68–42.20% and 41.52–47.35% compared with CK, respectively. PASP and CA showed the most significant improvements in the availability of Cd and Pb, respectively. In test B, the mass fractions of available Cd and Pb in rhizosphere soil were 1.078–1.170 mg·kg$^{-1}$ and 77.89–84.23 mg·kg$^{-1}$, respectively, which increased by 29.68–40.71% and 47.43–59.43% compared with CK. PASP increased the soil available Cd and Pb significantly.

#### 3.4.3. Effects of LMWOAs on Rhizosphere Soil pH and Nutrients at Maize Maturity Stage

It can be seen from Table 8 that different treatments reduced the soil pH to different degrees. Adding activated materials before maize planting reduced the soil pH by 0.03–0.10 units; adding activated materials at the ear stage reduced the soil pH by 0.08–0.14 units. The effect of spraying activator on soil pH acidification was greater in the growing and filling stages of maize. The concentrations of soil organic matter, total nitrogen, and

alkal-hydrolyzable nitrogen were significantly different under different treatments. The range of soil organic matter mass was between 20.60 and 23.34 g·kg$^{-1}$, which was higher than that of CK. The mass fractions of soil total nitrogen and alkali-hydrolyzed nitrogen ranged from 0.97 to 1.08 g·kg$^{-1}$ and from 70.11 to 79.40 mg·kg$^{-1}$. There was no significant difference in the soil available phosphorus and available potassium, and the mass fractions were 13.03–14.07 mg·kg$^{-1}$ and 116.67–23.50 mg·kg$^{-1}$, respectively.

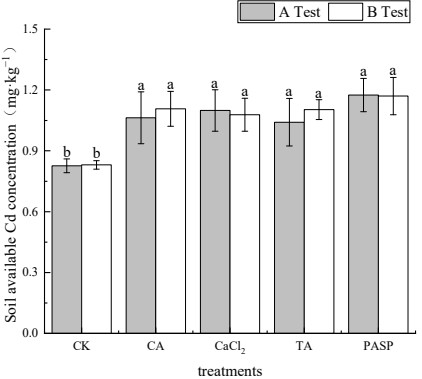 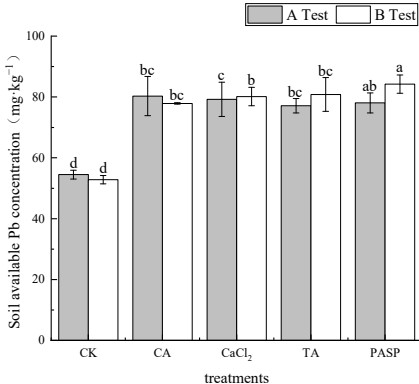

**Figure 3.** Effects of LMWOAs on the available Cd and Pb of heavy metals in rhizosphere soil at the maize maturity stage. Different small letters above the same column indicate significant difference between treatments ($p < 0.05$).

**Table 8.** Effects of LMWOAs on rhizosphere soil nutrients at the maize maturity stage.

| Test | Treatments | Soil pH | Organic Matter (g·kg$^{-1}$) | Total N (g·kg$^{-1}$) | Alkali-Hydrolyzed N (mg·kg$^{-1}$) | Available P (mg·kg$^{-1}$) | Available K (mg·kg$^{-1}$) |
|---|---|---|---|---|---|---|---|
| A | CK | 4.850 ± 0.053 ab | 19.83 ± 1.963 ab | 1.090 ± 0.020 a | 76.29 ± 2.182 ab | 13.62 ± 1.197 a | 118.7 ± 1.607 a |
| | CA | 4.753 ± 0.032 bc | 20.60 ± 1.006 ab | 1.080 ± 0.010 ab | 75.36 ± 2.457 ab | 14.07 ± 1.951 a | 118.5 ± 5.766 a |
| | CaCl$_2$ | 4.823 ± 0.055 abc | 19.99 ± 2.500 ab | 1.063 ± 0.068 ab | 70.11 ± 4.623 b | 13.37 ± 0.979 a | 121.0 ± 3.123 a |
| | TA | 4.793 ± 0.031 abc | 21.22 ± 2.174 ab | 1.043 ± 0.032 abc | 79.40 ± 2.605 a | 13.80 ± 0.885 a | 117.7 ± 5.508 a |
| | PASP | 4.820 ± 0.046 abc | 23.34 ± 2.116 a | 1.017 ± 0.078 abc | 70.53 ± 4.946 b | 9.00 ± 0.887 b | 118.3 ± 2.754 a |
| B | CK | 4.887 ± 0.038 a | 19.08 ± 0.393 b | 1.000 ± 0.036 bc | 76.56 ± 2.881 ab | 13.24 ± 1.115 a | 119.7 ± 1.258 a |
| | CA | 4.750 ± 0.046 abc | 20.45 ± 0.972 ab | 0.967 ± 0.015 c | 75.51 ± 4.117 ab | 13.95 ± 1.505 a | 116.7 ± 3.253 a |
| | CaCl$_2$ | 4.810 ± 0.036 abc | 20.30 ± 1.286 ab | 1.057 ± 0.032 ab | 76.76 ± 4.502 ab | 13.03 ± 0.340 a | 123.5 ± 2.291 a |
| | TA | 4.770 ± 0.072 bc | 21.45 ± 0.954 ab | 1.050 ± 0.010 ab | 75.69 ± 2.175 ab | 13.48 ± 0.858 a | 118.8 ± 1.258 a |
| | PASP | 4.800 ± 0.046 abc | 22.00 ± 1.368 ab | 1.007 ± 0.032 bc | 74.83 ± 4.045 ab | 13.55 ± 0.965 a | 117.8 ± 1.258 a |

Note: Different lowercase letters in the same column indicate a significant difference between treatments ($p < 0.05$).

## 4. Discussion

### 4.1. LMWOAs Enhances Heavy Metal Absorption, Enrichment, and Transport in Plants

Heavy metals in the soil with high durability and biodegradability and heavy metal ions can induce the plant secretion of organic acids via chelation reduce the toxicity of free metal ion activity in plants and organic acids as important ligands of heavy metals in plants and also involved in the absorption of heavy metals, transportation, storage, and other physiological metabolic processes [20]. Exogenous LMWOAs can chelate with Cd to form a mobile and soluble organic acid–Cd complex that can penetrate the lipid membrane of root cells, which is the main form of Cd uptake and collection in maize [21]. Through adsorption and precipitation, LMWOAs can have an affinity with the soil solid phase, promote the adsorption of heavy metals by plants, and reduce the concentration of heavy metals in soil [27]. The interaction of LMWOAs with soil organic anions, types, pH, and microorganisms can enable plants to efficiently adsorb heavy metals in polluted soil [28]. The concentration of the carboxyl group involved in the soil adsorption reaction of citric acid is high, which has a high adsorption affinity. It has been found that the scientific application of 1-hydroxyethylidene-1,1-diphosphonic acid (HEPD) and D-glucuronic acid (D-GA) can

promote the absorption of cadmium and lead in polluted soil by solanum nigrum [29]. Previous studies have shown that exogenous LMWOAs can enhance the absorption of heavy metals in soil by plants. For example, applying 3 mmol·kg$^{-1}$ citric acid can improve the absorption of Zn, Cu, Pb, Ni, and Cd by *Liriope platyphylla* Wang et Tang, and the repair index is between 60% and 187% [22]. The application of 2 mmol·kg$^{-1}$ citric acid promoted the accumulation of heavy metals in the stalks and roots of *Iris halophila* Pall [23]. The application of LMWOAs such as acetic acid, oxalic acid, citric acid, malic acid, and tartaric acid can improve the uptake of Cd by the shoots of Brassica napus, with an increase rate of more than one time [25]. The combined application of citric acid and Kocuria rhizophila can promote the accumulation of Cd, Cr, Cu, and Ni in the shoots of *Glycine max* L. [30]. In this study, exogenous LMWOAs can improve the Cd concentrations of maize kernels, cores, straws, and roots in the range of 9.26–53.92%, 0.35–9.48%, 28.49–56.85%, and 33.65–121.18%, respectively, compared with CK. Exogenous LMWOAs can improve the Pb concentration of maize kernels, cores, straws, and roots in the ranges of 3.40–34.73%, 4.60–28.09%, 24.56–61.39%, and 10.47–69.01% respectively, compared with CK. It is consistent with the previous research results. The availability of heavy metals in soil and plant biomass significantly effects the application and development of phytoremediation technology [31], and the levels of biological enrichment coefficient, transport coefficient, and extraction coefficient in phytoremediation are closely related to the plant biomass [32]. The scientific application of LMWOAs can improve the transport efficiency of Cd in the roots and shoots of *Phytolacca americana L.*, change the morphological distribution of Cd in soil, improve the activation effect, and strengthen the bioaccumulation of Cd in plants [33]. Sebastian et al. [34] found that, after the application of citric acid, the BCF of Cd in the upper part of Sedum Sedum increased by 54.51%, the BCF of Cd in the root system decreased by 62.53%, the TF of Pb increased by 66.67%, and the TF of zinc increased by 73%, which significantly improved the extraction efficiency of heavy metals from Sedum Sedum. The exogenous application of 5 mmol·kg$^{-1}$ malate increased the BCF, RE, and TF of *Celosia argentea* L. by three, two, and one times, respectively [35]. After spraying 10 mmol·kg$^{-1}$ limonic acid, the BCF, RE, and TF of *Solanum nigrum* L. increased by 19.7%, 23.3%, and 0.3%, respectively [29]. The application of limonic acid increased the uptake and accumulation of Cd by 3 times and 2.3 times in the roots and branches of Festuca arundinacea, respectively [32]. The exogenous application of 1 mmol·kg$^{-1}$ acetic acid could significantly increase the Cd concentration in the roots of *Brassica napus* L. and enhance the enrichment effect [25]. The application of appropriate concentrations of citric acid and tartaric acid can improve the absorption capacity of Cd and the transport capacity of Cd from roots to shoots of solanum [36]. In this study, after adding 10 mmol·L$^{-1}$ LMWOAs, the BCF of maize straws and roots Cd increased by 28.39–61.74% and 33.68–121.09%, respectively, and the BCF of maize straws and roots Pb increased by 24.47–61.12% and 10.66–68.73%, respectively, which is consistent with the results of previous studies. Therefore, the scientific application of LMWOAs can promote the absorption of heavy metals in plants and strengthen the enrichment effect of plants.

### 4.2. LMWOAs Increase the Availability of Soil Heavy Metals and Change Soil Properties

The induced stress of heavy metals in soil can destroy the pH stability of plant cell fluid and use Pb to increase LMWOA secretion in rhizosphere soil and reduce the pH of plant rhizosphere [37]. In this study, after the addition of activators such as LMWOAs, the pH values of maize rhizosphere soil at the mature stage decreased to different degrees, with the decrease ranging from 0.03 to 0.14 units. Mostly because the plant production of LMWOAs exists mainly in the form of free in the cytoplasm and in the form of the anion release into the rhizosphere, it results in the cytoplasm of the plant in an ionic imbalance of yin and yang; in order to balance the process, the hydrogen ions in the form of a proton pump discharge from plant cells into rhizosphere soil and reduce the rhizosphere pH [38], promote the release of heavy metals from the soil, improve the solubility of heavy metals, increase the mobility of heavy metals, and enhance the process of heavy metal absorption

and transfer [39]. In this study, different treatments could promote the activities of available soil heavy metals Cd and Pb by 28.68–42.20% and 41.52–59.43%, respectively. Although activators can activate heavy metals in soil, improve the availability of heavy metals, enable them to be absorbed by plants, and improve the repair efficiency of plants, activators will be gradually decomposed in the soil environment over time, reducing their activation effect. Relevant studies have found that, when biodegradable activators are applied to heavy metal-contaminated soil, their activation effect on heavy metals decreases with the extension of the time treatment, and the half-life is about 5–10 days. More studies have shown that naturally degradable chelators, such as EDDS, have a half-life of only about 2.5 days. In this study, the effect of the addition of liquid-activated material on the absorption of maize plants was significantly better in the ear stage when maize was growing strongly than that before maize planting, which may be due to the fact that the LMWOAs used in this study had a natural degradable effect on the soil environment and a short half-life. Therefore, the addition of a liquid activator to the rhizosphere during the most vigorous ear period of maize growth could significantly promote the uptake of heavy metals in the activated soil by maize plants. The application of exogenous LMWOAs can also effectively change the properties of rhizosphere soil and promote the dissolution of unavailable minerals in soil [40]. Relevant studies have shown that LMWOAs can provide soil with carbon sources and activated nutrients, such as available phosphorus and alkali-hydrolyzed nitrogen, which can significantly increase the soil productivity [41]. In this study, the concentration of soil organic matter has been improved to different degrees, and the concentrations of soil organic matter, total nitrogen, and alkali-hydrolyzed nitrogen have significant differences, which is similar to the results of previous studies. However, there was no significant difference in the soil available P and available K concentrations, indicating that adding an activator had no significant effect on the soil available P and available K.

*4.3. LMWOAs Promote Plant Nutrient Uptake and Growth*

The reduction of plant biomass under heavy metal stress is an irreversible plant growth inhibition phenomenon [35]. Roots are the main organs for plants to absorb heavy metals from soil, and heavy metals will cause harm to roots and inhibit plant growth [42]. LMWOAs are mainly produced by plant rhizosphere release, soil microbial synthesis, and plant litter degradation [43]. They participate in the process of plant growth and development and can promote plant growth by providing the available phosphorus and available iron compounds [44], producing depolymerizing humus, activating auxin, and alleviating stress damage to photosynthetic organs [45]. Lam et al. found that the microbial inoculants helped the plants to grow healthily and increased the yield [46]. Farid et al. showed that citric acid could resist biotic and abiotic stresses and enhance nutrient absorption to promote plant growth [47]. Li et al. found that oxalic acid and citric acid could enhance the respiration of ramie (*Boehmeria nivea* L.) by improving the root activity, promoting the absorption of nutrients and increasing the plant fresh weight and dry weight by 44.6% and 74.4%, respectively [24]. By adding tartaric acid, Tao et al. found that tartaric acid promoted the dissolution of carbonate, thereby increasing the aboveground biomass of Sedumalfredii by 15% [48]. In this study, adding LMWOAs and $CaCl_2$ increased the maize biomass by 1.18–6.32%, which was the same as the results of previous studies. Studies have shown that an exogenous application of 5 $mmol \cdot L^{-1}$ citric acid under chromium stress can increase the dry weight of roots, stems, and leaves of Helianthus annuus by 31%, 23%, and 42% and the fresh weight by 21%, 28%, and 32%, respectively [47]. The application of citric acid, tartaric acid, and oxalic acid under copper stress can significantly promote the biomass of *Castorcommunis* L. [49]. The application of exogenous citrate and malate under Cd stress can increase the biomass of rice (*Oryza sativa* L.) by 119.0% [39]. The application of 2 $mmol \cdot kg^{-1}$ citric acid could increase the shoot biomass and root biomass of Iris halophila Pall by 42.8% and 51.6%, respectively [33]. However, if the concentration of LMWOAs is too high, it may cause a significant Pb decline in plant biomass. Li et al.

found that the application of excessive citric acid could have a negative effect on the root morphology of Seduma Seduma, reducing its root length, rhizome, and root volume [50]. In this study, it was found that the addition of 10 mmol·L$^{-1}$ PASP before planting reduced the whole biomass of maize by 0.44%. This may be because the high concentration of LMWOAs exceeded the tolerance range of the plants, which damaged their benign growth state, increased the inhibition of the development of lateral roots near the root tip of plants, and may even have increased the necrosis of the root tip cells [51]. Therefore, different types and concentrations of LMWOAs have significantly different effects on plant growth and development and biomass. The scientific application of LMWOAs under heavy metal stress can effectively regulate plant development and significantly improve plant biomass.

## 5. Conclusions

The study took LMWOAs as the research object to study the effect of different LM-WOAs on enhancing the extraction and removal of heavy metals from soil by maize. The results showed that LMWOAs could increase the yield and biomass of maize and enhance the absorption of heavy metals cadmium and lead in the soil by maize. The enriched heavy metals were mainly distributed in the nonedible parts of maize. Although the transport of Cd and Pb to maize grains increased, maize grains could be used as raw materials for animal feed according to GB 13078-2017 in China. Under the conditions of 2.41 mg·kg$^{-1}$ Cd-contaminated soil and 203.41 mg·kg$^{-1}$ Pb-contaminated soil, the three LMWOAs can strengthen the removal of heavy metals from maize, and the comprehensive effect of CA is the most significant. Although maize is not the most ideal variety for plant extraction, the results of the study can be generalized due to the large concentrations of Cd and Pb in the nonedible parts of maize. Further research should focus on the treatment and application of the nonedible parts of maize.

**Author Contributions:** Conceptualization, Z.M. and H.L.; methodology, W.Y.; software, X.Z.; validation, R.T., Y.M. and H.H.; formal analysis, J.H.; investigation, C.C.; resources, J.Z.; data curation, X.Z.; writing—original draft preparation, R.T.; writing—review and editing, R.T.; visualization, W.Y.; supervision, Y.M.; project administration, H.H.; funding acquisition, Z.M. All authors have read and agreed to the published version of the manuscript.

**Funding:** This research was funded by the National Key R&D Program Project "Technology Demonstration of Cadmium Arsenic Pollution Prevention and Ecological Security in Urban Rural Areas" (No. 2018YFD0800203) and the Anhui Provincial Science and Technology Major Research Project "Development and Application of Efficient Nanoremediation Materials for Heavy Metal Pollution in Farmland" (No. 17030701053).

**Institutional Review Board Statement:** Not applicable.

**Informed Consent Statement:** Informed consent was obtained from all subjects involved in the study.

**Data Availability Statement:** The data presented in this study are available on request from the corresponding author.

**Conflicts of Interest:** The authors declare no conflict of interest.

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
