# Peer review of "Effect of LMWOAs on Maize Remediation of Cadmium and Plumbum Pollution in Farmland"

_sustainability, doi:10.3390/su142114580_

Round 1
Reviewer 1 Report
This study provided a study on the phytoextraction by maize as assisted by natural chelators. Although many studied have been reported in this research filed, this study conducted with a field trial but not pot experiment, which may be some new compared with previous studies. However, there are many flaws in this MS, especially for the calculation. At present, it should be not accepted before the authors corrected them.
Abstract:
(1) Please give the full name of “LMWOAs” when it present in the first time of the abstract part.
(2) Please unify the name of “corn” and “maize”.
(3) Please simplify the abstract.
Introduction Part:
(1) Authors presented the advantage of Natural chelators, but why to choose these three organic chelators should be clarified.
Methods part:
(1) The total amount of the liquid materials should be provided for Test A and B treatments.
Results part:
(1) Please add the units in Table 2 and 3.
(2) Table 5 title: “CD-contaminated”; “kg/mu”? mu is not a international unit.
(3) As Table 4 “BCF=plant element content (mg·kg-1)/soil element content (mg·kg-1).” Since different Cd concentration accumulated in different parts, how to calculate? For example, the corn, straw and root Cd is 0.292, 0.840 and 0.701 mg·kg-1, respectively, and the soil Cd is 2.41 mg·kg-1. How to calculate as the result that “the CK of BCF for Cd is 0.853” ?
(4) Table 5, the calculation is still ambiguous. How many maize planted in ha? As 0.078 mg/plant and 60000 plant/ha (assumed), then the total Cd is 4.7 g/ha but not 313 g/”mu”.
(5) In the Results part or whole-MS, all of the “content” is “concentration” in fact, the meaning of these two words is different. Furthermore, fig.3 “effective Cd” is no meaning and should be “available Cd”.
Discussion part
(1) “4.2. LMWOAs increased soil heavy metal activity”, Generally, “activity” is point to the enzyme but not Cd.
(2) In fact, maize is not a optimal plant species for phytoextraction, because of it’s low assimilation for Cd and being a food crop. However, the fact that Cd accumulation in root and straw is much higher than the grains may make it be a logical story. Authors should pay more attention on it in this part. Furthermore, the disposal of Cd-contaminated biomass after harvest should be considered.
Author Response
Dear reviewer, it is a great honor to be recognized by you, and thank you for your valuable suggestions on this manuscript. Next, the author will give you an effective reply to your suggestions on revision. The revised part of this manuscript has been marked in red font.
Abstract:
(1) Please give the full name of “LMWOAs” when it present in the first time of the abstract part.
The author has given the full name when low molecular organic acids first appeared in the abstract, and added (LMWOAs).
- Please unify the name of “corn” and “maize”.
The author has unified the name as "maize".
(3) Please simplify the abstract.
The author has simplified the abstract while retaining the main content.
Introduction Part:
- Authors presented the advantage of Natural chelators, but why to choose these three organic chelators should be clarified.
The author has added "the three kinds of LMWOAs belong to the category of natural chelating agent, CA and TA belong to carboxylic acid compounds, PASP belongs to compounds. The three kinds of LMWOAs are biodegradable, relatively friendly to the environment, and are not easy to cause secondary pollution. In the past pot experiments, the extraction effect is significant, and there is a strong remediation potential." in the introduction to explain the advantages and differences of these three low molecular organic acids.
Methods part:
- The total amount of the liquid materials should be provided for Test A and B treatments.
The author has added the molar mass of each solid material and the amount of liquid used in each cell in the test treatment section. Please check.
Results part:
- Please add the units in Table 2 and 3.
The author has added units in Table 2 and Table 3.
- Table 5 title: “CD-contaminated”; “kg/mu”? mu is not a international unit.
The author has revised the title of Table 5, and all the "mu" units in the article have been converted into the international unit "hectare"
- As Table 4 “BCF=plant element content (mgkg-1)/soil element content (mg·kg-1).” Since different Cd concentration accumulated in different parts, how to calculate? For example, the corn, straw and root Cd is 0.292, 0.840 and 0.701 mg·kg-1, respectively, and the soil Cd is 2.41mg·kg-1. How to calculate as the result that “the CK of BCF for Cd is 0.853” ?
Thank the reviewers for finding this problem. It is unreasonable to add the element content of different parts to calculate the element content of the whole plant without the support of relevant literature. This problem is caused by the author's failure to consult more literature when writing. At present, the enrichment factor of the whole plant has been modified to the enrichment factor of corn straw and root, and the discussion part has been modified. Please refer to it.
- Table 5, the calculation is still ambiguous. How many maize planted in ha? As 0.078 mg/plant and 60000 plant/ha (assumed), then the total Cd is 4.7 g/ha but not 313 g/”mu”.
We are also grateful to the reviewers for finding this problem. This problem is that the author nominates "g" as "kg" when labeling units, which has been corrected.
- In the Results part or whole-MS, all of the “content” is “concentration” in fact, the meaning of these two words is different. Furthermore, fig.3 “effective Cd” is no meaning and should be“available Cd”.
According to the opinions of experts, the author has modified the full text of "content" to "concentration" and the "effective Cd" in Figure 3 to "available Cd"
Discussion part
- “2. LMWOAs increased soil heavy metal activity”, Generally, “activity” is point to the enzyme but not Cd.
Thanks for the reminder of the reviewers, the "activity" in 4.2 has been modified to "concentration"
- In fact, maize is not a optimal plant species for phytoextraction, because of it’s low assimilation for Cd and being a food crop. However, the fact that Cd accumulation in root and straw is much higher than the grains may make it be a logical story. Authors should pay more attention on it in this part. Furthermore, the disposal of Cd-contaminated biomass after harvest should be considered.
Thank you very much for the valuable opinions put forward by the experts. The author has strengthened the analysis and discussion of this part and enriched the content of the article.
The above is the author's reply to the suggestions made by the reviewers. This manuscript may still have some deficiencies, and we hope that the experts can give advice.
Reviewer 2 Report
1- Please reduce your article abstract section.
2- Please revise your article key words, choose suitable words.
3- Please mention country name instead of our country throughout the manuscript.
4- Please correct subscript and superscript.
5- Please clearly mention your objectives and novelty.
6- Please improve the language of the current version.
7- Please revise your equations in proper format.
8- Please bold the letters on the error bars of all the figures.
9- What do you think that all of your added treatments can act as chelator?
10- please remove numbering from your conclusion section.
Author Response
Dear reviewer, thank you very much for your approval of this article. The following author will give you a positive response based on your comments and suggestions. The revised part of the manuscript has been marked in red.
- Please reduce your article abstract section.
According to the simplified summary you proposed, the author simplified the summary without changing the original meaning, making it more intuitive.
- Please revise your article key words, choose suitable words.
The author adjusted and added the keywords according to the changes you proposed.
- Please mention country name instead of our country throughout the manuscript.
According to what you mentioned, the author has added the objective, the shortcomings of previous research and the improvement of this article in the foreword.
- Please correct subscript and superscript.
According to your suggestions, the author has corrected the superscripts and subscripts that do not comply.
- Please clearly mention your objectives and novelty.
According to what you mentioned, the author has added the objective, the shortcomings of previous research and the improvement of this article in the foreword.
- Please improve the language of the current version.
According to your requirements, the author has improved the language expression of the full text.
- Please revise your equations in proper format.
According to your request, the author has modified the format of the equation.
- Please bold the letters on the error bars of all the figures.
According to your request, the author has bold the title blocks of all tables in the article.
- What do you think that all of your added treatments can act as chelator?
According to your question, the author clearly puts forward the effect of all methods in the discussion and conclusion section.
- please remove numbering from your conclusion section.
According to your request, the author has refined the conclusion and deleted the number.
Reviewer 3 Report
Although this research was well edited based on author(s)'s original research results, this research article should have to revise before publication as one of research articles in international journal(s).
[Major reversion]
1) 3 page(P), 15 line(L)
According to China's "Soil Environment Standard Standard for Soil Pollution~" --->Soil Environment Standards for Soil Pollution~
What is this? Do you think that it is a research article which was submitted for peer-review?
2) Objective(s)
Author(s) need to consider that why this research is important on the viewpoint of globalization not just one country. Then author(s) should be more specific and apparent objective(s) in the main contents of this research article.
3) Results
So what? Where and how can we apply using the results of this research?
Author(s) should have to consider the applicability of this research's results, and then revise the main contents of this research.
4) References
If possible, please use references within 5 years.
Author(s) should have to use at least one or more research article(s) in the journal, "Sustainability".
[Minor reversion]
6) Number(s) and unit expression
All number(s) one blank Unit except %
Ex.: 20 cm
7) Effective numbers
Please use same effective numbers in a same research article(including Table or Figure).
Please use effective numbers at least 3 or more.
Ex.: 4.67+/-0.000, 42.8=+/-3.44, 153+/-5.60
Author Response
Dear manuscript reviewer, the author is very honored to be recognized by you for this manuscript. At the same time, according to your valuable comments, the author also recognizes the shortcomings. The author has revised the manuscript according to your requirements for revision. Please check it. The revised part of the manuscript has been marked in red.
[Major reversion]
1) 3 page(P), 15 line(L)
According to China's "Soil Environment Standard Standard for Soil Pollution~" --->Soil Environment Standards for Soil Pollution~
What is this? Do you think that it is a research article which was submitted for peer-review?
According to your suggestions, the author has revised the standards cited in this article again to make them more complex in English.
2) Objective(s)
Author(s) need to consider that why this research is important on the viewpoint of globalization not just one country. Then author(s) should be more specific and apparent objective(s) in the main contents of this research article.
According to your suggestions, the author has made a relatively detailed revision to the foreword, reviewed the previous international and domestic research achievements, refined the purpose and innovation of the article, and revised it in the article.
3) Results
So what? Where and how can we apply using the results of this research?
Author(s) should have to consider the applicability of this research's results, and then revise the main contents of this research.
According to your point of view, the author re summarized the results and proposed the promotion and application of the research results in a more detailed and clear way.
4) References
If possible, please use references within 5 years.
Author(s) should have to use at least one or more research article(s) in the journal, "Sustainability".
According to your suggestions, the author has used the references of the past five years as much as possible. Some of the literature authors believe that it is of reference value to this study, so they keep it. At the same time, they also cited articles in the journal, "Sustainability".
[Minor reversion]
5) Number(s) and unit expression
All number(s) one blank Unit except %
Ex.: 20 cm
The author has revised all figures and unit expressions into the required format.
6) Effective numbers
Please use same effective numbers in a same research article(including Table or Figure).
Please use effective numbers at least 3 or more.
Ex.: 4.67+/-0.000, 42.8=+/-3.44, 153+/-5.60
The author has modified all the figures in this article and reserved 4 significant figures.
Round 2
Reviewer 1 Report
As the soil Cd is 2.41 mg/kg, and the 20 cm surface soil/ha is 2250000 kg/ha, then the total amount of remediated (or extracted) Cd is (2.41-0.3)*2250000/1000=4747.5 g/ha.
As the table showed the extracting amount for CK is 4.683 g/ha, so the needed remediated time is 1073 year. Let alone the fact that the extracting efficiency could not always maintain this level during long-term remediation.
As I mentioned in the first comments, the remediation efficiency of maize is much lower than hyperaccumulators. In fact, it belongs to a very low Cd accumulating crop species. If planting rice in such high Cd-contaminated soil (2.41 mg/kg), the rice grain may accumulate higher than 1.0 mg/kg Cd. Thus, it is no meaning to calculate the relative repair life. Authors should delete this content of MS, and put more effort to explain the advantage of root and shoots for Cd remediation in discussion part. This issue only mentioned in the conclusion part is not acceptable.
Author Response
Comments and Suggestions for Authors
As the soil Cd is 2.41 mg/kg, and the 20 cm surface soil/ha is 2250000 kg/ha, then the total amount of remediated (or extracted) Cd is (2.41-0.3)*2250000/1000=4747.5 g/ha.
As the table showed the extracting amount for CK is 4.683 g/ha, so the needed remediated time is 1073 year. Let alone the fact that the extracting efficiency could not always maintain this level during long-term remediation.
As I mentioned in the first comments, the remediation efficiency of maize is much lower than hyperaccumulators. In fact, it belongs to a very low Cd accumulating crop species. If planting rice in such high Cd-contaminated soil (2.41 mg/kg), the rice grain may accumulate higher than 1.0 mg/kg Cd. Thus, it is no meaning to calculate the relative repair life. Authors should delete this content of MS, and put more effort to explain the advantage of root and shoots for Cd remediation in discussion part. This issue only mentioned in the conclusion part is not acceptable.
Thank you very much for your valuable comments and suggestions on the manuscript again. As you mentioned in the article, compared with the hyperaccumulation plant, maize is not the best choice for plant extraction, but maize has the characteristics of large biomass. The concentration of heavy metals that can be enriched in straws and roots is far higher than that in grain. Therefore, the author discussed the absorption and enrichment of maize straws and roots in the discussion section, and cited previous research literature as a reference, At the same time, the relative repair life in the text and table is deleted. Please check and correct.
Reviewer 3 Report
I think that this revised research article may be suitable for the publication of one of research articles in the journal, "Sustainability".
Author Response
Comments and Suggestions for Authors
I think that this revised research article may be suitable for the publication of one of research articles in the journal, "Sustainability".
Thank you very much for reviewing this manuscript again. It is a great honor to be recognized by you for this manuscript. Sustainability is a very influential journal. This manuscript can not be published in Sustainability without your suggestions and corrections. Thank you again.
Round 3
Reviewer 1 Report
In the revised content of discussion part:
1. abbreviating "zinc, cadmium..."
2. ”The availability and biomass of heavy metals in soil ...” what is meaning?
3. compare the calculations between total amount of grain Cd and total amount of shoot, root Cd of this study.
Author Response
Thank you again for your valuable comments. It is hard to imagine that without your modification, the article might not be published. I will reply one by one according to your suggestions.
In the revised content of discussion part:
- abbreviating "zinc, cadmium..."
These elements have been abbreviated according to your requirements.
- ”The availability and biomass of heavy metals in soil ...” what is meaning?
This sentence is the author's description of the problem.Has been modified to "The availability of heavy metals in soil and plant biomass significantly affect the application and development of phytoremediation technology ".
- Compare the calculations between total amount of grain Cd and total amount of shoot, root Cd of this study.
The comparative analysis has been carried out, and the expression of the article has been changed to "In this study, exogenous LMWOAs can improve the Cd concentration of maize kernels, cores, straws and roots in the range of 9.26% to 53.92%, 0.35% to 9.48%, 28.49% to 56.85% and 33.65% to 121.18% respectively compared with CK. Exogenous LMWOAs can improve the Pb concentration of maize kernels, cores, straws and roots in the range of 3.40% to 34.73%, 4.60% to 28.09%, 24.56% to 61.39% and 10.47% to 69.01% respectively compared with CK".